# Effect of Powder on Tribological and Electrochemical Properties of Nylon 66 and Ultra-High Molecular Weight Polyethylene in Water and Seawater Environments

**DOI:** 10.3390/polym13172874

**Published:** 2021-08-27

**Authors:** Wanxing Xu, Tian Yang, Shengpeng Zhan, Dan Jia, Lixin Ma, Saisai Ma, Haitao Duan

**Affiliations:** 1Wuhan Research Institute of Materials Protection, Wuhan 430030, China; xu_wanxing@126.com (W.X.); shanxiyt@163.com (T.Y.); sp_zhan@sina.com (S.Z.); jiadan0510@163.com (D.J.); mlx19950316@163.com (L.M.); 15733295024@163.com (S.M.); 2State Key Laboratory of Special Surface Protection Materials and Application Technology, Wuhan Research Institute of Materials Protection, Wuhan 430030, China

**Keywords:** polymer, electrochemical, wear

## Abstract

Polymer materials are used increasingly in marine machinery and equipment; their tribological properties and effect on the water environment have garnered significant attention. We investigate the effect of water or seawater environment containing powder on tribology and electrochemistry of polymer materials. A friction test involving nylon 66 (PA66) and an ultrahigh molecular weight polyethylene (UHMWPE) pin–disc (aluminum alloy) is performed in seawater or water with/without polymer powder, and the solution is analyzed electrochemically. The results show that the tribological properties of the UHMWPE improved by adding the powder to the solution, whereas the PA66 powder demonstrates abrasive wear in a pure water environment, which elucidates that the synergistic effect of powder and seawater on UHMWPE reduces the wear, and the synergistic effect of pure water and powder aggravates the wear. The results of electrochemical experiments show that after adding powder in the friction and wear tests, the powder can protect the pin by forming a physical barrier on the surface and reducing corrosion, and the changes are more obvious in seawater with powder in it. Through electrochemical and tribological experiments, the synergistic effect of solution environment and powder was proved.

## 1. Introduction

Owing to the depletion of land resources and the gradual discovery of marine resources in recent years, ocean exploration has increased worldwide [1,2,3,4]. Additionally, the development of sea-related mechanical materials and equipment has garnered increasing attention [5,6]. Polymers and their composite materials are widely used in seals, piston rings, valves, lubricating bearings, and bearing retainers for sea-related equipment, owing to their superior properties [7,8]. In particular, polymers and their composites have proven to be promising seawater-lubricated materials because of their excellent corrosion resistance, low friction, and high wear resistance under seawater lubrication [9,10,11,12,13].

The friction and wear processes of polymers in water environments have been investigated extensively [3,14,15]. Some studies have focused on the effects of pressure, running speed, marine environment, and other factors on the tribological properties of polymers [16,17,18]. Liu [13] investigated the relationship between hydrostatic pressure and the wear behavior of thermoplastic polymers sliding in seawater. It was demonstrated that an increase in the hydrostatic pressure increased the wear rate of the polymer. Some researchers have investigated the effects of salinity and particle concentration on friction and corrosion using electrochemical methods. Chen et al. discovered that the corrosion rate of Inconel 625 alloy was the highest in 3% seawater owing to the synergistic effect between salinity and dissolved oxygen [19]. Although studies have shown that the friction between polymers and steel in NaCl solution produces rust, the role of wear debris in the friction and wear process has not been investigated [20]. A few researchers improved the corrosion resistance and wear resistance of surfaces via coating. Ye investigated the effect of epoxy resin content in a coating on performance improvement and discovered that 0.5 wt% graphene oxide/epoxy specimen displayed better lubrication properties than other groups [21]. Polymer materials can be used to fabricate artificial joints. Therefore, some researchers have investigated the wear mechanism, wear state, and other properties of polymer materials by analyzing the existing state, particle size distribution, and agglomeration of debris accumulated in the lubricating fluid; however, this approach has not been applied to mechanical research [22]. Debris is an important friction product, which cannot disperse and participate in physical and chemical reactions in the atmospheric environment as well as in the solution environment. Studies regarding debris in water environments are scarce, and most of them do not associate debris with tribological properties and its effect on the solution [22,23,24]. 

Hence, this study was performed to determine the effects of debris from friction and wear tests on water and seawater environments. Because it is difficult to make a large amount of debris, we use powder instead of grinding chips for preliminary study. In electrochemical experiments, the similarity can be confirmed. In this study, a friction test involving a pin (PA66, UHMWPE) and a disc (aluminum alloy) was performed in seawater or a pure aqueous solution containing polymer powder in order to simulate equipment such as water-lubricated bearings. After performing the friction tests, scanning electron microscopy (SEM) and confocal scanning optical microscopy were performed to investigate the wear mechanism of the polymer in an aqueous environment with wear debris, and the three-dimensional surface topography of the wear scar was characterized to obtain its width and depth. The findings of this empirical study will provide insights into the mechanism of polymer wear debris in seawater and water environments, thereby enriching the tribological theory of polymers.

## 2. Materials and Methods

### 2.1. Friction and Wear Tests

PA66 bars (Dong Xing Plastic Factory, Wuxi, China) and UHMWPE bars (Cheng da insulating plastic factory, Suzhou, China) were processed into pins measuring 8 mm × 20 mm; their properties are listed in Table 1. The flatness of pins and discs was 0.03 mm, the parallelism was 0.03 mm, the dimensional tolerance was 0.04 mm, and the processing method was lathe machining. Aluminum alloy is one of the most widely used non-ferrous metal structural materials in the industry. It offers good corrosion resistance and has been widely used in aviation, aerospace, automobile, machinery manufacturing, shipbuilding, and chemical industries. Therefore, aluminum alloy 1a90 was selected as the friction pair material, and the chemical composition of aluminum 1a90 is shown in Table 2. Aluminum plates with a thickness of 10 mm and a diameter of 50 mm were used in the friction pair, and these 1A90 discs were polished to a surface roughness in the range of Ra 0.2 μm and Ra 0.35 μm. An aluminum disc/UHMWPE (pin) friction pair and an aluminum disc/PA66 (pin) friction pair were used to perform friction and wear experiments. A contact schematic diagram of the frictional couple is shown in Figure 1. The processing method of powder was liquid nitrogen crushing, the particle size of the powder was 150 mesh (0.1 mm), and the concentration of the polymer powder was 8 g/L; the powder was dispersed in the solution in the form of agglomeration. The seawater used in the experiment was simulated seawater composed of coral salt, the element composition of which is presented in Table 3. As shown in Figure 1, to obtain the surface in full contact, the disc was fixed on the spindle and rotated at a speed of 1000 r/min. The pin was fixed on the fixture, and the positive force exerted by the lever was 30 N, with a rotation radius of 21 mm. The environment temperature was 25 °C, and the friction test was conducted for 3600 s.

After alcohol cleaning and drying, the surface morphology and roughness after friction were scanned and analyzed using a Micromeasure2 confocal scanning optical microscope (Sciences et Techniques Industrielles de la Lumière, Aix en Provence, France). The scanning area was 3 mm × 3 mm, and the step size was 20 μm × 20 μm. After ultrasonic acetone cleaning and drying, the samples were sprayed with gold, and the micromorphology of the friction surface was observed and analyzed using SEM.

### 2.2. Electrochemical Experimental Parameters

The test instrument used in the experiment was a Princeton electrochemical workstation (Partstat 2273, Ametek, Berwyn, PA, USA). The tests were conducted in a conventional three-electrode cell; a Pt plate and a saturated calomel electrode were employed as the counter electrode and reference electrode, respectively. Blocks measuring 10 mm × 10 mm × 3 mm were used as the operating electrodes. They were embedded in a Teflon holder to expose an area measuring 1 cm^2^ to the solution. Prior to the test, each sample was sanded using 1000-mesh sandpaper and cleaned with alcohol, and the scanning rate was 0.5 mv/s. Subsequently, the self-corrosion potential, polarization, and curve were measured. When measuring the self-corrosion potential, the scanning frequency was set within 10 kHz–100 MHz.

## 3. Results

### 3.1. Effect of Powder on Friction and Wear

The wear amount of polymer pins was measured by confocal scanning optical microscopy. As shown in Figure 2, compared with the specimen in the solution without powder, the wear amount of the UHWMPE pin in the solution with powder was lower. The experiments in Figure 2 were repeated three times. Despite the effect of water absorption, the amount of wear reduced significantly after the powder was added to the solution. In addition, the abrasion loss of the UHMWPE in seawater was significantly lower than that in water, which is because the corrosion products of seawater play a role in protecting the surface. Figure 3 shows the change in the friction coefficient with time when the UHMWPE sample was tested on the disc friction and wear tester. It is evident that in the seawater and pure water environments, adding UHMWPE powder makes the friction coefficient of the material comparatively lower. After adding powder into pure water, the average wear amount decreased from 0.57045 to 0.3269 mm^3^. After adding powder into seawater, the average wear amount decreased from 0.27435 to 0.1963 mm^3^. The three-dimensional topography presented in Figure 4 shows that the wear mark on the pin surface was lighter in the seawater and the water containing powder. The roughness in Figure 4 was measured and listed in Table 4. It can be seen that the roughness becomes smaller after adding powder.

Figure 5 shows surface micrographs at 500× magnification obtained via SEM for UHMWPE specimens after the friction and wear tests. As shown, in the water environment, the surface of the UHMWPE specimen exhibited significant abrasive wear, and the width of the wear scar was similar to the particle size of the powder. By contrast, in other groups, indications of abrasive wear were absent, but lighter or heavier furrows were observed. Some other researchers also discovered that when UHMWPE was rubbed against steel, its molecular chains became dissociated and free radicals chelated with the counter face [25,26].

In order to verify the conjecture, the powder adheres to the surface, the content of carbon and oxygen in the wear scar of the aluminum disc was detected and compared with that on the aluminum disc without experiment. Compared with Figure 6 and Table 5, it can be seen that after adding the powder, the powder obviously adheres to the disc, and the carbon content at the wear mark increases evidently.

To further confirm the viewpoint that the powder adheres to the friction surface, the infrared spectrum of the aluminum plate was investigated, and the results are shown in Figure 7. Compared with the typical Poly tetra fluoroethylene (PTFE) peaks at 2919 and 2854 cm^−1^ in the C–H region of the raw composite, two clear peaks at 2848 and 2922 cm^−1^ as well as a COO- peak at 1456 cm^−1^ were observed on the steel surface [25], indicating that the UHMWPE chains were dissociated and underwent a number of reactions generating fluorinated carboxylic acids to chelate with both the steel counterpart and the reinforcing filler [27,28,29]. The infrared spectrum shows that the absorption peak of C–H appeared in the infrared spectra of three aluminum plates matched with UHMWPE, indicating that matter transfer occurred in these experiments. The polymer material was transferred and attached to the aluminum disc during the friction process, thereby reducing the wear of the aluminum disc. However, compared with the strong absorption peaks in other studies [27,28,29,30,31], the intensity of the polymer characteristic peaks on the steel surface measured in this experiment was weak, indicating a relatively small amount of material transfer.

The UHWMPE pin in seawater containing powder was the group that exhibited the least wear in all experiments. Additionally, the white-light Figure shows that the wear mark on the pin surface was the lightest. Some products were clearly shown in the infrared spectrum. These results illustrate that UHWMPE is a suitable material for marine machinery and equipment and that it can yield better tribological properties in seawater.

The friction and wear behaviors of PA66 exhibited different rules. Figure 8 shows that the abrasion loss of the PA66 pin increased after adding powder into the solution, and the average wear amount raised from 1.179 mm^3^ to 1.8475 mm^3^ after adding powder into pure water, the average wear amount increased from 0.45235 mm^3^ to 1.09075 mm^3^ after adding powder into seawater. Changes in the friction coefficient with time are presented in Figure 9. As shown, after adding powder to the water, the friction reduction performance deteriorated significantly. However, the friction coefficient of the PA66 pin with the addition of the powder was smaller than that in the seawater without powder, which is due to PA66 powder is not easy to adhere to the surface, both abrasive wear and friction film occurred. The lubrication of seawater also reduced the friction coefficient, and powder added in solution increased the wear. Figure 10 illustrates serious abrasive wear of PA66 pin occurred in water environment with powder, which differs from the finding for the UHMWPE pins.

Combining with Figure 10 and Figure 11, it can be seen that serious abrasive wear of PA66 pin occurred in a pure water environment with powder. There are some furrows on the surface of other samples. After adding the powder into the seawater, the wear marks become lighter. This is consistent with the decrease in the friction coefficient. The roughness in Figure 10 was measured and listed in Table 6. It can be seen from Table 6 that the roughness becomes larger after adding powder in the water environment and reduced in the seawater environment.

In order to find out whether the adhesion phenomenon of powder on the surface of the aluminum disc also occurs in the experiment of PA66, we studied the aluminum disc grinded with PA66 pin, and the results are shown in Figure 12 and Table 7. No particles can be detected on the aluminum plate after adding powder in the water environment, but the powder can be detected on the grinding parts in the seawater environment. These results indicate that the powder cannot adhere to the aluminum disc in the water environment but can adhere to the aluminum disc in the seawater environment.

In the infrared spectrum of the aluminum disc matched with PA66 (Figure 13), the typical absorption peak of amide I caused by C=O stretching vibration was observed at 1643 cm^−1^, and the absorption peak of amide II was caused by N–H stretching vibration appeared at 1535 cm^−1^. This implies that in the seawater, the powder shielded the friction surface and therefore reduced friction and wear [25,32,33]. The peak was not observed for the sample in water containing powder. This is because PA66 powder does not impose a lubricating effect in water but simulates abrasive wear. Additionally, this result was confirmed by the three-dimensional topography shown in Figure 10.

For PA66, the most severe friction and wear occurred in water containing powder, whereas the least severe occurred in seawater. As shown in Figure 13, compared with the other three solutions, the water containing powder did not yield a friction film, which may be due to the higher hardness of the PA66 powder and the properties of the solution, which renders it difficult to form a friction film on the surface.

After adding powder, the tribological properties of UHMWPE and PA66 show a big difference. UHMWPE powder can enhance the antifriction and wear resistance of the pin, which may be due to the low hardness of UHMWPE powder and better compatibility with the matrix. In this test, the Rockwell hardness of UHMWPE powder is 52, and that of PA66 powder is 108. The hardness of PA66 powder is higher, which is easier to cause abrasive wear rather than forming friction film.

The wear amount of the two materials in seawater is less than that in pure water, indicating that seawater has a good lubrication effect on the two polymer materials. After adding powder, the friction coefficient and wear of the UHMWPE pin decreased, indicating that the powder played an antifriction role. After adding powder, the wear of PA66 increased, and SEM photos also showed obvious scratches. However, compared with the reduction of friction coefficient of the UHMWPE pin after powder addition, the friction coefficient of PA66 pin decreases only when the powder is added to seawater. These results show that the synergistic effects of different powders and solutions are also different.

UHMWPE material has self-lubricating property, while PA66 material does not. Generally speaking, the self-lubricating performance is reflected in the friction process of the base material. The powder cannot disperse and flow in the air as in water. Adding powder to the solution and dispersing the powder in the aqueous solution is equivalent to strengthening the lubrication effect of self-lubricating materials and causing abrasive wear for materials that cannot be self-lubricating. Solution and powder have effects on the friction and wear of materials, respectively, and the solution can be used as the medium for powder dispersion in the environment. This reflects the synergistic effect of powder and solution environment.

### 3.2. Effect of Powder on Solution Corrosivity

Figure 14 shows that, compared with the solution prior to the friction test, the polarization curves of the PA66 and UHMWPE samples in seawater shifted to the left. Furthermore, a significant difference is indicated between the anode and cathode polarization curves. Before corrosion occurred, the anodic polarization curve was steep, the cathodic polarization curve was short and gentle, and the anode polarization curves were extremely unbalanced. This shows that the copper electrode in the corrosion system exhibited great resistance during anode dissolution and that the passivation film on the copper surface was intact and passivated [34,35,36]. By contrast, the polarization curves of the PA66 and UHMWPE samples in water shifted to the right after the friction test, the polarization region of the anode was narrow, and the stability of the passivation film might not be as stable as that formed in seawater.

As shown in Table 8 and Table 9, in a water and seawater environment, wear debris is generated after friction test, and corrosion current density is reduced, indicating that the conductivity of the solution is reduced [37]. However, this does not apply to pure water experiments. The corrosion current density of pure water is the lowest because it contains almost no salt. However, in pure water solution containing powder, the corrosion current density after the experiment is lower than that before the experiment. In the experiment of adding powder, the decrease in the corrosion current density of the two materials is larger, which illustrates that the addition of the powder and the generation of wear debris in the solution can also protect the electrode. In all experiments, the corrosion current density of PA66 in pure water plus powder environment decreased the least. In addition, the decrease in the corrosion current is more obvious in seawater. The results of electrochemical experiments are consistent with those of tribological experiments, which implies that the improvement of corrosion resistance is positively correlated with the improvement of antifriction performance, and there is a certain relationship between friction and corrosion. In other words, the electrochemical reaction also exists in the friction process.

A few studies have reported the relationship between friction and corrosion and that electrochemical corrosion affects friction and wear mechanisms [3,38,39,40]. The experimental results suggest that the powder and debris on the surface of the friction pair formed a barrier. Compared with PA66, UHMWPE powder demonstrated better self-lubricating performance and reduced friction wear; meanwhile, the PA66 specimen in the water environment demonstrated the least adsorption. Therefore, it exhibited abrasive wear owing to the high hardness after adsorption on the surface. The electrochemical law is similar to the law of friction and wear, indicating that there is a certain interaction between electrochemistry and friction and wear.

The corrosion current density decreased more obviously after adding powder. The powder was dispersed in the solution so that it constitutes both a medium and a reactant; the polymer materials participate in reactions in more than one form. The addition of powder changed the properties of the solution and reduced the conductivity of the solution, especially in seawater because the better lubricating properties of seawater allowed the powder to adhere to the electrode surface and reducing the corrosion current density, which shows the synergistic effect of powder and solution environment.

It can be seen from the previous paper that whether the wear debris can form a friction film and improve the friction and wear resistance of the matrix mainly has two factors: one is affected by the properties of the material itself, and the other is the solution environment. Hence, we propose a primary model and present the possible wear mechanism in Figure 15 and Figure 16, where the connection between C and O is based on the study of Przedlacki [27].

## 4. Conclusions

In this study, we used powder to simulate the effects of polymer debris on the friction and wear processes as well as the electrochemical parameters of solutions. A pin-on-disc contact tribometer was used to investigate the tribological behavior of a polymer pin matched with an aluminum plate. An electrochemical workstation was used to test the electrochemical parameters of the solution prior to and after the friction and wear tests.

Generally speaking, the self-lubrication we mentioned refers to the properties of the matrix material. However, in this paper, we realize self-lubrication in a new way by dispersing powder in a solution environment and combine this self-lubrication with the lubrication of the solution to prove its synergy. The results obtained were as follow:(a)Adding powder decreased the wear volume and friction coefficient of the UHMWPE specimen. After adding powder into pure water, the average wear amount decreased from 0.57045 to 0.3269 mm^3^. After adding powder into seawater, the average wear amount decreased from 0.27435 to 0.1963 mm^3^. For the PA66 specimens, the addition of powder increased the wear amounts of all samples, but adding powder increased the coefficient of the pin in the water environment and decreased that in the seawater environment. After adding powder into pure water, the average wear amount increased from 1.179 to 1.8475 mm^3^. After adding powder into seawater, the average wear amount increased from 0.45235 to 1.09075 mm^3^.(b)The results of tribological experiments indicated that wear debris can improve the tribological properties of UHMWPE in water and seawater environments by shielding the friction surface as well as reduce wear. However, the wear debris shielding the surface of the PA66 pin could not reduce wear in the water environment, the powder in the water stimulated abrasive wear, and the friction reduction and wear resistance of PA66 in the water environment decreased. This is due to the high hardness of PA66 powder, resulting in abrasive wear.(c)The results of electrochemical experiments demonstrated that powder can form a physical barrier on the surface and reduce the corrosion current to protect the material. In UHMWPE tests, the corrosion current density of seawater decreased from 6.342 to 2.199 μA/cm^2^ before and after friction test, and from 11.530 to 2.083 μA/cm^2^ in seawater with powder, reflecting the synergistic effect of seawater and powder. In PA66 tests, the corrosion current density of PA66 decreased from 6.342 to 3.286 μA/cm^2^ before and after tribological test in seawater, and from 10.420 to 3.229 μA/cm^2^ in seawater with powder. Moreover, in the salt-containing seawater solution, the powder is easier to adhere to the surface, and the law of friction reduction and corrosion resistance of the powder is similar, which indicates that the interaction of electrochemistry, friction, and wear occur. It also shows the synergistic effect of powder and solution environment.

## Figures and Tables

**Figure 1 polymers-13-02874-f001:**
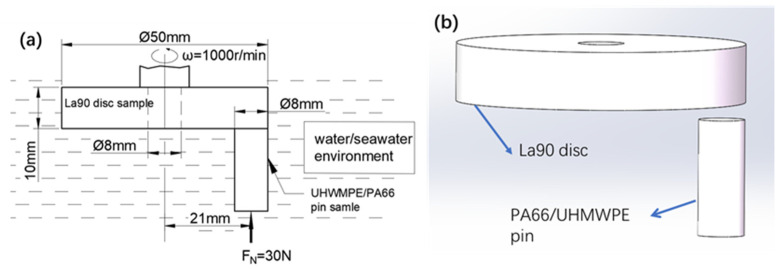
Contact diagram of aluminum alloy disc/UHMWPE (PIN): (**a**) Two-dimensional graph; (**b**) three-dimensional graph.

**Figure 2 polymers-13-02874-f002:**
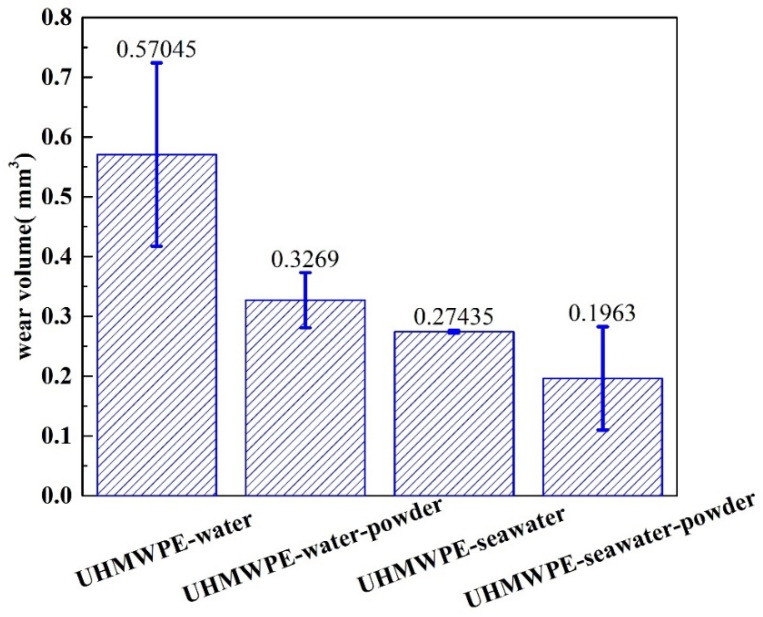
Wear amount of UHMWPE pins tested using pin–disc friction and wear tester. The error bar in the Figure represents the standard deviation.

**Figure 3 polymers-13-02874-f003:**
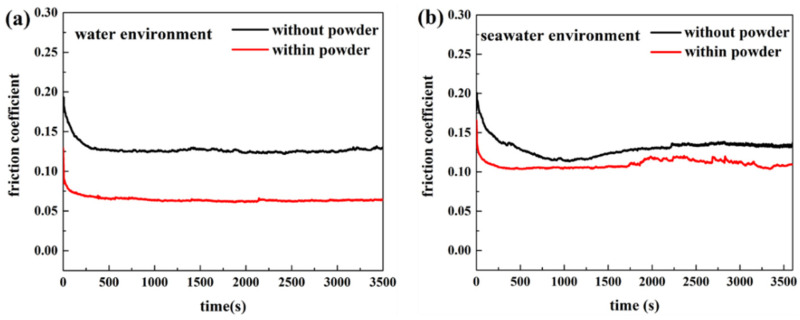
The friction coefficient of the UHMWPE pin in (**a**) water and (**b**) seawater environments.

**Figure 4 polymers-13-02874-f004:**
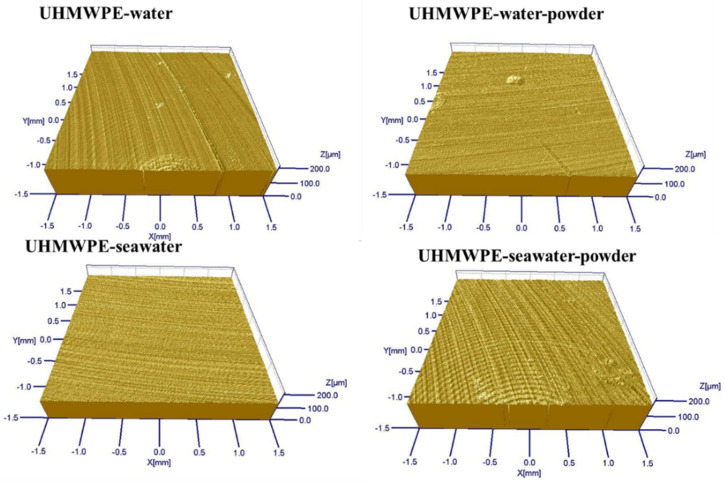
Micrograph images captured using white-light interference three-dimensional profilometer.

**Figure 5 polymers-13-02874-f005:**
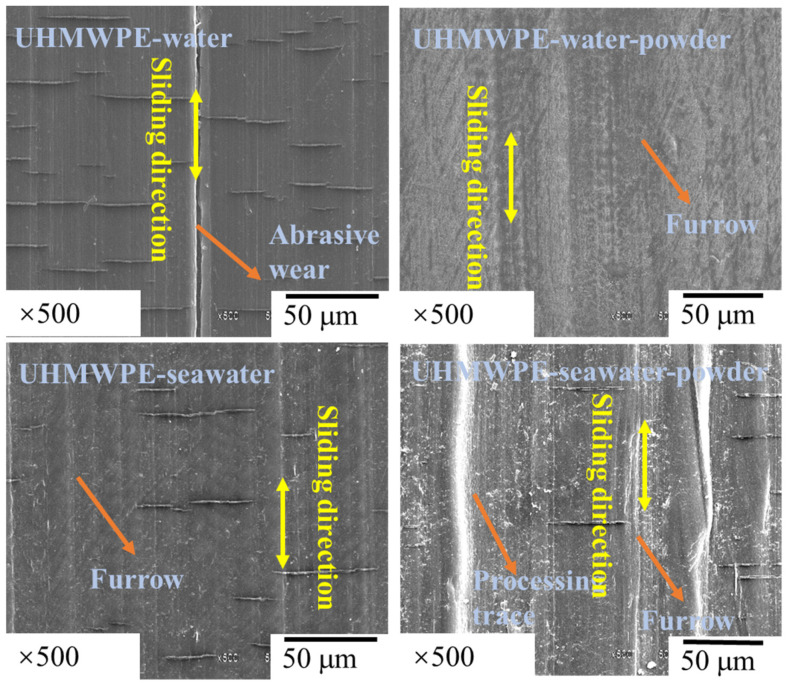
SEM images of UHMWPE pins.

**Figure 6 polymers-13-02874-f006:**
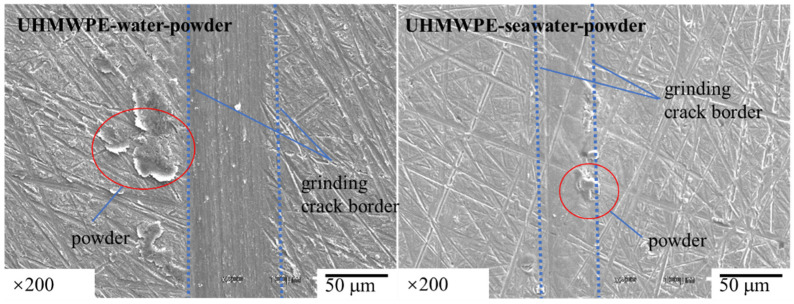
SEM images of wear marks on 1a90 aluminum plate matched with UHMWPE pins in solution with powder.

**Figure 7 polymers-13-02874-f007:**
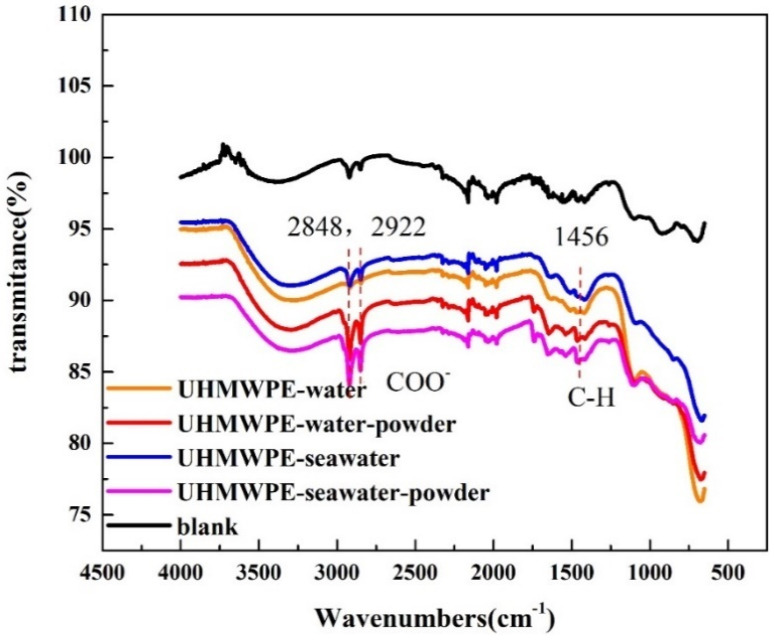
Infrared spectrogram of aluminum surface of grinding components.

**Figure 8 polymers-13-02874-f008:**
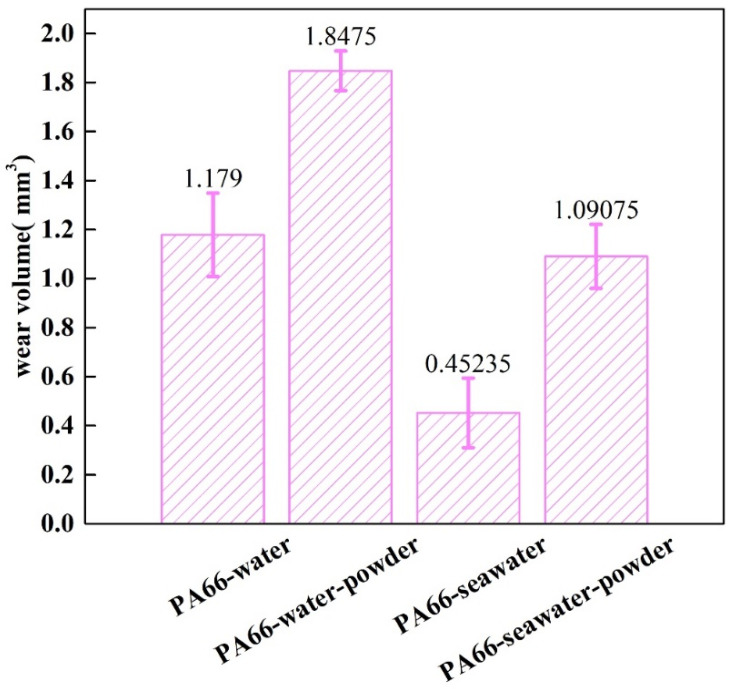
Wear amounts of PA66 pins tested using pin–disc friction and wear tester. The error bar in the Figure represents the standard deviation.

**Figure 9 polymers-13-02874-f009:**
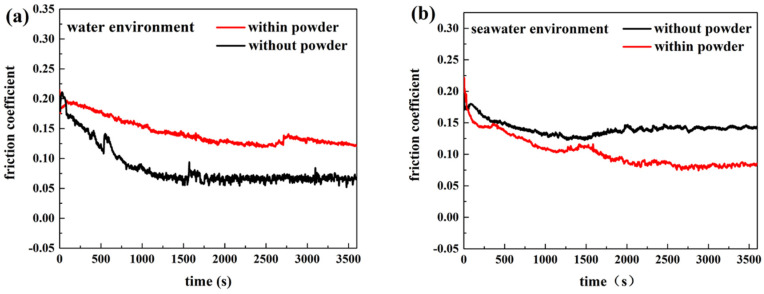
The friction coefficient of PA66 pin in (**a**) water and (**b**) seawater environments.

**Figure 10 polymers-13-02874-f010:**
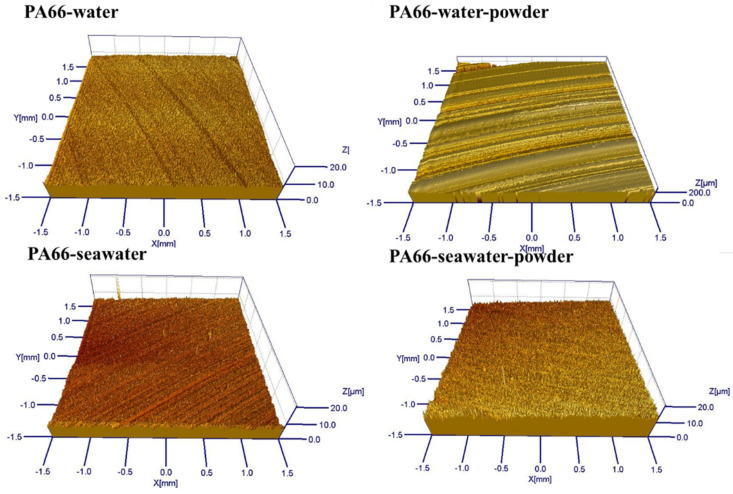
Micrograph images captured using white-light interference three-dimensional profilometer.

**Figure 11 polymers-13-02874-f011:**
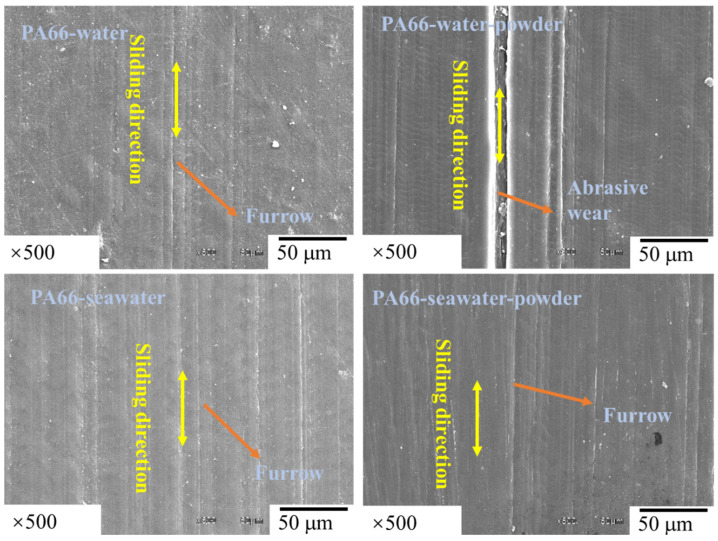
SEM images of PA66 pins.

**Figure 12 polymers-13-02874-f012:**
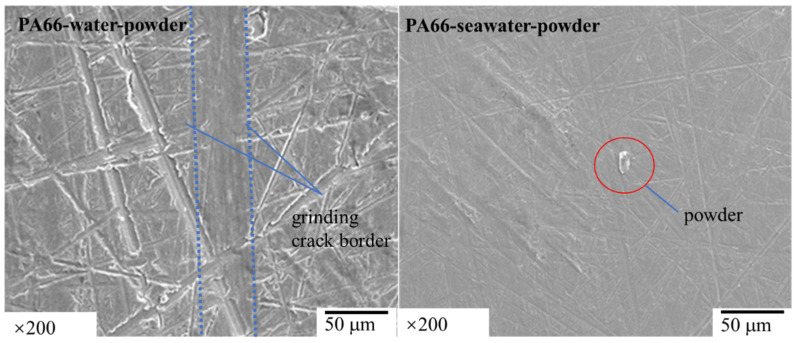
SEM image of wear marks on the 1a90 aluminum plate matched with PA66 pins in solution with powder.

**Figure 13 polymers-13-02874-f013:**
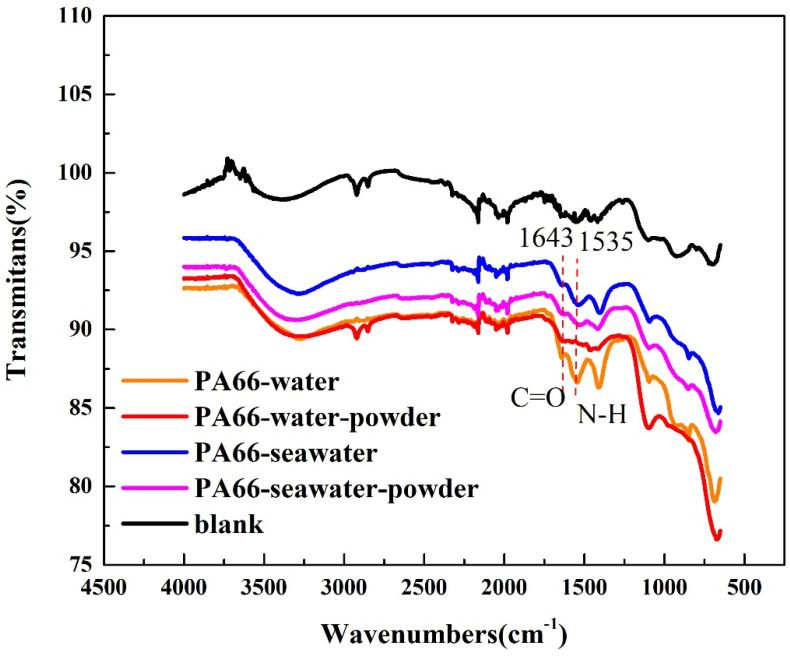
Infrared spectrogram of the aluminum surface of grinding components.

**Figure 14 polymers-13-02874-f014:**
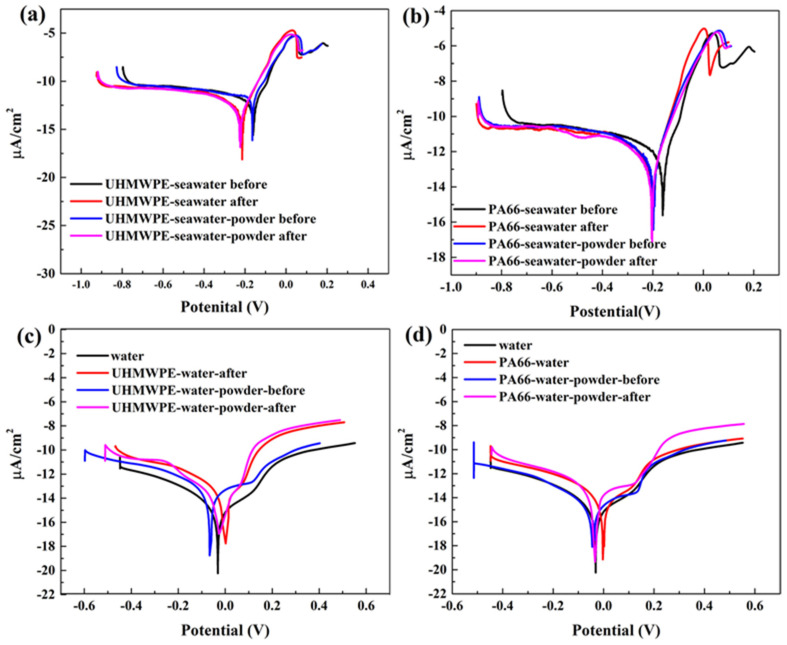
Polarization curve measured using electrochemical workstation: (**a**) UHMWPE pin in seawater; (**b**) PA66 pin in seawater; (**c**) UHMWPE pin in water; (**d**) PA66 pin in water.

**Figure 15 polymers-13-02874-f015:**
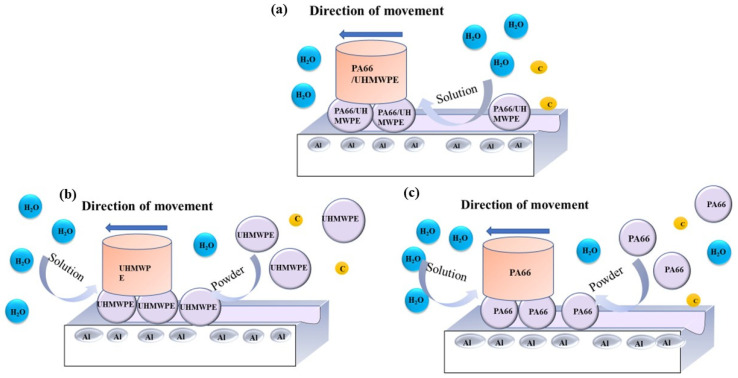
Suggested schematic diagram of lubrication layer formation. (**a**) UHMWPE/PA66 pins in water; (**b**) UHMWPE pins in water with powder in it; (**c**) PA66 pins in water with powder in it.

**Figure 16 polymers-13-02874-f016:**
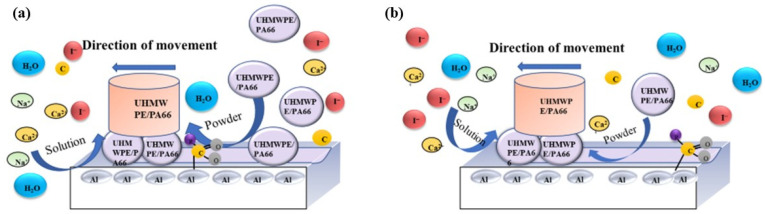
Suggested schematic diagram of lubrication layer formation.(**a**) seawater; (**b**) seawater with powder in it.

**Table 1 polymers-13-02874-t001:** The properties of PA66 and UHMWPE.

	Density(g/cm^3^)	Tensile Strength (MPa)	Elongation at Exercise (%)
PA66	1.26	37	3.8
UHMWPE	0.93	22	>50

**Table 2 polymers-13-02874-t002:** Chemical composition of aluminum 1a90.

Element	Si	Fe	Cu	Mg	Zn	Mn	Ti	v
Mass fraction	0.02	0.02	0.01	0.01	0.01	0.01	0.005	0.005

**Table 3 polymers-13-02874-t003:** The composition of artificial seawater.

Element	Ca^2+^	Mg^2+^	I^3-^	K^+^
Concentration(mg/L)	410	1200	0.01	360

**Table 4 polymers-13-02874-t004:** The roughness of the UHMWPE pin surface.

Pin-Solution Environment	UHMWPE -Water	UHMWPE -Water-Powder	UHMWPE -Seawater	UHMWPE -Seawater-Powder
Ra(nm)	280	234	350	302

**Table 5 polymers-13-02874-t005:** Element analysis of wear marks on the aluminum plate.

Pin Solution Environment	Empty	UHMWPE-Water	UHMWPE- Water-Powder	UHMWPE- Seawater	UHMWPE -Seawater-Powder
C	0.22	1.72	2.14	1.61	3.16
O	1.11	5.36	4.85	52.96	26.76

**Table 6 polymers-13-02874-t006:** The roughness of PA66 pins surface.

Pin-Solution Environment	PA66 -Water	PA66 -Water-Powder	PA66 -Seawater	PA66 -Seawater-Powder
Ra(nm)	146	4311	181	275

**Table 7 polymers-13-02874-t007:** Element analysis of wear marks on the aluminum plate against PA66 pins.

Pin-Solution Environment	Empty	PA66 -Water	PA66 -Water-Powder	PA66 -Seawater	PA66 -Seawater-Powder
C	0.22	1.73	1.12	1.78	2.75
O	1.11	5.1	19.45	13.22	7.91

**Table 8 polymers-13-02874-t008:** Comparison of the corrosion current density with and without UHMWPE powder before and after friction and wear tests.

Solution Environment	Corrosion Current Density(μA/cm^2^)
seawater	6.342
seawater after friction test	2.199
seawater with powder before friction test	11.530
seawater with powder after friction test	2.083
water	0.704
water after friction test	1.214
water with powder before friction test	4.425
water with powder after friction test	0.463

**Table 9 polymers-13-02874-t009:** Comparison of the corrosion current density with and without PA66 powder before and after friction and wear tests.

Solution Environment	Corrosion Current Density(μA/cm^2^)
seawater	6.342
seawater after friction test	3.286
seawater with powder before friction test	10.420
seawater with powder after friction test	3.229
water	0.704
water after friction test	1.728
water with powder before friction test	1.709
water with powder after friction test	1.046

## Data Availability

Not applicable.

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
