# Peer review of "Effect of Powder on Tribological and Electrochemical Properties of Nylon 66 and Ultra-High Molecular Weight Polyethylene in Water and Seawater Environments"

_polymers, 2021, doi:10.3390/polym13172874_

Round 1
Reviewer 1 Report
The article is valuable because the test results can provide information about the friction and wear processes of polyamide PA66 and polyethylene PE UHMW during sliding on aluminium alloy in a water or seawater environment. These are interesting practical applications of polymeric materials.
The only weakness can be poor statistical information about the results of experimental research. There is no information about error bars presented in Fig.2, Fig. 8. Are they standard deviations or confidence intervals? Please include a short comment in the text of the article
In addition, Figure 15 is not readable (Font too small). Maybe it is better to set the schematic diagram vertically instead of horizontally?
Author Response
The article is valuable because the test results can provide information about the friction and wear processes of polyamide PA66 and polyethylene UHMWPE during sliding on aluminium alloy in a water or seawater environment. These are interesting practical applications of polymeric materials. The only weakness can be poor statistical information about the results of experimental research. There is no information about error bars presented in Fig.2, Fig. 8. Are they standard deviations or confidence intervals? Please include a short comment in the text of the article In addition, Figure 15 is not readable (Font too small). Maybe it is better to set the schematic diagram vertically instead of horizontally?
Thanks so much for your sincere comments. We add explanations of the error bars, they mean standard deviation, and we split Figure 16 into two figures. Thank you for your support and encouragement.
Reviewer 2 Report
The authors have performed an interesting (and surprising) study of the effect of powder on the tribological and electrochemical performance of PA66 adn UHMWPE. I have some hesitations about the relevance of this study, but I believe that the novelty of this research can balance my decision to accept after minor revisions.
I only have one technical comment. I think that an analogous study with a metallic pin and plastic disks might be interesting. Have the authors thought about it?
There are several typos that should be changed:
Table 1: cm3 and Mpa.
Line 111: cm2
Lines 166 and 167: cm-1
The authors use disc and disk alternatively in the manuscript. Just pick one.
Author Response
Reviewer 2’s comments
The authors have performed an interesting (and surprising) study of the effect of powder on the tribological and electrochemical performance of PA66 adn UHMWPE. I have some hesitations about the relevance of this study, but I believe that the novelty of this research can balance my decision to accept after minor revisions. I only have one technical comment. I think that an analogous study with a metallic pin and plastic disks might be interesting. Have the authors thought about it?
We appreciate your suggestion earnestly. The reason why the metal disc and plastic pin are selected is that the disc is rotating and the pin is stationary. This friction pair simulates sliding guide rails, water lubricated bearings of polymer materials, etc. According to your suggestions, we makes further supplementary explanations in Line 80, and further study the rotating plastic disc and metal pin in the follow-up work.
2 There are several typos that should be changed: Table 1: cm3 and Mpa. Line 111: cm2 Lines 166 and 167: cm-1 The authors use disc and disk alternatively in the manuscript. Just pick one.
Thank you so much for your comments. We are very sorry for the mistakes in this manuscript and inconvenience they caused in your reading. In the revised manuscript, the typo errors of “cm3” and “Mpa” have been corrected to “cm3” and “MPa”, respectively.We think the word "disc" is more accurate and has been replaced “disk” with “disc”. Thank you for your reminder.
